# Hybrid Unilamellar Vesicles of Phospholipids and Block Copolymers with Crystalline Domains

**DOI:** 10.3390/polym12061232

**Published:** 2020-05-29

**Authors:** Yoo Kyung Go, Nurila Kambar, Cecilia Leal

**Affiliations:** Department of Materials Science and Engineering, University of Illinois at Urbana−Champaign, Urbana, IL 61801, USA; go6@illinois.edu (Y.K.G.); nkambar2@illinois.edu (N.K.)

**Keywords:** giant hybrid vesicles, phospholipids, diblock copolymer, semi-crystalline polymer, phase-separation

## Abstract

Phospholipid (PL) membranes are ubiquitous in nature and their phase behavior has been extensively studied. Lipids assemble in a variety of structures and external stimuli can activate a quick switch between them. Amphiphilic block copolymers (BCPs) can self-organize in analogous structures but are mechanically more robust and transformations are considerably slower. The combination of PL dynamical behavior with BCP chemical richness could lead to new materials for applications in bioinspired separation membranes and drug delivery. It is timely to underpin the phase behavior of these hybrid systems and a few recent studies have revealed that PL–BCP membranes display synergistic structural, phase-separation, and dynamical properties not seen in pure components. One example is phase-separation in the membrane plane, which seems to be strongly affected by the ability of the PL to form lamellar phases with ordered alkyl chains. In this paper we focus on a rather less explored design handle which is the crystalline properties of the BCP component. Using a combination of confocal laser scanning microscopy and X-ray scattering we show that hybrid membranes of 1,2-dipalmitoyl-sn-glycero-3-phosphocholine (DPPC) and methoxy-poly(ethylene glycol)-*b*-poly(ε-caprolactone) (mPEG-*b*-PCL) display BCP-rich and PL-rich domains when the BCP comprises crystalline moieties. The packing of the hydrophilic part of the BCP (PEG) favors mixing of DPPC at the molecular level or into nanoscale domains while semi-crystalline and hydrophobic PCL moieties bolster microscopic domain formation in the hybrid membrane plane.

## 1. Introduction

Phospholipids (PLs) and amphiphilic block copolymers (BCPs) both have the ability to self-assemble into a variety of nanostructures [1,2], the lamellar phase being one of the most prominent mesophases. This has led to the development of polymersomes [3]—closed membrane systems analogous to liposomes that have many advantages in the applications of drug delivery [4]. Despite similarities in their assembled structures, PL and BCP membranes have rather distinct properties including membrane thickness, bending and stretch moduli [5], as well as local and translational molecular dynamics [6]. In addition, while PL chemistry is rather limited, BCPs can be synthesized with a plethora of desired functional groups. The prospect of stabilizing a hybrid membrane comprising PLs and BCPs—and concomitantly a *hybridosome* particle is tantalizing as it can harness the advantages of liposomes and polymersomes and their use in biomedical applications including controlled drug delivery, artificial cell development, and biosensors [7,8]. In order to develop such materials, it is important to study the self-assembly, dynamical, and phase-behavior properties of hybrid PL–BCP membranes. In the recent years, hybrid PL–BCP membranes have indeed emerged [6,7,9,10,11,12] as materials where the combined properties are not merely an additive of those observed in pure components. For example, the dynamics of alkyl chain conformational changes in 1,2-dipalmitoyl-sn-glycero-3-phosphocholine (DPPC) membranes are enhanced at room temperature by hybridization with poly(butadiene)-*b*-poly(ethylene oxide)—PBD-PEO BCPs [6,12]. In addition, DPPC membranes in the gel-phase (L_β_) hybridized with PBD-PEO in a giant unilamellar vesicle (GUV) exhibit the formation of large phase-separated PL-rich and BCP-rich domains [13]. Controlling the crystallinity of the PL alkyl chains have a significant effect on the ability of hybrid membranes of PC lipids and PBD-PEO to mix at the molecular level or to phase separately [14]. Clearly, controlling the molecular packing of the PL and polymer constituents such as (1) the phase/physical state of PL membranes (gel-phase—L_β_, liquid disordered—L_α_, or liquid ordered—L_o_) [9,15,16], (2) hydrophobic mismatch between PL and polymer [9], as well as (3) architecture/form of polymers (graft copolymer or block copolymer) [16] is expected to have a significant impact on the structure, dynamics, and phase separation behavior of hybrid membranes [7,8,17].

Herein, we highlight a new perspective focusing on the phase/physical state of the BCP by choosing a polymeric system that has the ability to crystallize at room temperature. Pure crystalline polymer [18]/BCP [19] systems can assemble into particles and have been recently described as *crystalsomes.* In these systems, the crystallizable polymer consists of poly(L-lactic acid) (PLLA or PLA) and assembles into rigid nanoparticles in a size of ca. 200 nm in diameter. In the work presented here, we investigate the structure, phase behavior, and molecular order in hybrid membranes comprising DPPC and a BCP that can crystallize at room temperature - methoxy-poly(ethylene glycol)-*b*-poly(ε-caprolactone (mPEG-*b*-PCL). PCL is a semi-crystalline hydrophobic block with a molecular weight of 5000 g/mol and mPEG is the amorphous hydrophilic group with a molecular weight of 2000 g/mol leading to a weight fraction of 0.29 (*f*_PEG_ = 0.29). As a function of weight fraction and molecular weight, (m)PEG-*b*-PCL can form a variety of morphologies such as precipitates, spheres, vesicles, and worm-like micelles in aqueous solution [20,21,22,23]. However, crystallization alters the general self-assembly behavior due to changes in membrane elasticity, curvature, and hydration state [22]. DPPC is in the *gel-phase* (L_β_) at room temperature reaching membrane rigidity values (ca. 350 k_B_*T*) comparable to most BCP systems (35–400 k_B_*T*). In their fluid, liquid disordered state—L_α_ membrane bending rigidity is around 20 k_B_*T* [17]. It is noteworthy that hybrid membranes comprising mPEG-*b*-PCL and DPPC have been investigated previously [24,25,26]. The Demetzos group investigated the interaction of mPEG-*b*-PCL(5.25k*-b-*2.25k g/mol) (*f*_PEG_ of 0.70) with DPPC in the form of “chimeric” nanovesicles (<1 μm) and the main finding was that steric effects of mPEG-*b*-PCL chains induce the formation of hybrid vesicles smaller than expected [24,25]. Kang et al. [26] very recently used PEG-*b*-PCL-*b*-PEG triblock copolymers to synthesize giant hybrid unilamellar vesicles (GHUVs) comprising DPPC. The focus of that work was to investigate the shape and stability of the GHUVs. In our work, we focus on elucidating the assembly process and phase-separation behavior of a hybrid system where both BCP and PL are able to form domains with a high degree of molecular order and crystallinity.

Using confocal laser scanning microscopy (CLSM) we could, for the first time, directly observe the formation of PL-rich and BCP-rich phase-separated domains at room temperature in hybrid DPPC–mPEG-*b*-PCL GHUVs. Domains are readily visible by using standard PL labeling techniques while the BCP domains could be imaged by covalently tagging mPEG-*b*-PCL with fluorescein isothiocyanate (FITC). We determined that mPEG-*b*-PCL with a weight fraction of hydrophilic block (*f*_PEG_) of 0.29 and the molecular weight of 2k-*b*-5k g/mol enables the formation of bilayered GHUVs with DPPC. While pure mPEG-*b*-PCL vesicles display irregular membranes consistent with distorted PCL crystalline domains, hybridization with DPPC alleviates that constraint. Exposing a pre-formed GHUV to a hypertonic environment leads to enhanced phase separation of the membrane into PL-rich and BCP-rich domains as well as additional distortions of the GHUV membrane. Small Angle X-ray Scattering (SAXS) and Wide Angle X-ray Scattering (WAXS) support the existence of nano and microscopic domain formation in DPPC–mPEG-*b*-PCL hybrid membranes.

## 2. Materials and Methods

### 2.1. Materials

1,2-Dipalmitoylphosphatidylcholine (DPPC or 16:0 PC) and 1,2-dipalmitoyl-sn-glycero-3-phosphoethanolamine-N-(lissamine rhodamine B sulfonyl) (ammonium salt) (16:0 Liss Rhod PE or Rhod B PE) were purchased from Avanti Polar Lipids (Alabaster, AL, USA). Methoxy poly(ethylene glycol)-*block*-poly(ε-caprolactone) (mPEG-*b*-PCL, 2k-*b*-5k g/mol) diblock copolymer, fluorescein 5(6)-isothiocyanate (FITC), and anhydrous N,N-Dimethylformamide (DMF) were purchased from Sigma-Aldrich (St. Louis, MO, USA). All chemicals and solvents were used without additional purification.

### 2.2. FITC-labelling of mPEG-b-PCL

FITC-labelled block copolymers, (mPEG-*b*-PCL)-FITC, were prepared by conjugation chemistry of the hydroxyl end group of the PCL block in mPEG-*b*-PCL and the isothiocyanate group of FITC [27]. Fluorescein 5(6)-isothiocyanate (FITC) (1 mg) was reacted with mPEG-*b*-PCL (10 mg) in 2 mL of anhydrous DMF for 48 h at room temperature. After the reaction, the product was collected and dried by rotary evaporator.

### 2.3. Giant Unilamellar Vesicle (GUV) formation by Electroformation

For pure BCP GUV formation, 5 μL of a (mPEG-*b*-PCL)-FITC solution (25 mg/mL, chloroform) was mixed with 2 μL methanol, then was spread onto an indium tin oxide (ITO)-coated glass slide (70–100 Ω/sq) from Sigma-Aldrich (St. Louis, MO, USA). The solvent was dried for 1 day at room temperature in vacuum. For the GUV formation, we constructed a polydimethylsiloxane (PDMS) chamber (W 1 cm X H 2 cm X D 3 mm) with two ITO-coated glass slides [28]. PDMS was made by using the SYLGARD™ 184 Silicone Elastomer Kit from Dow (Midland, MI, USA). The base and curing agent were mixed in 10:1 ratio by weight. The swelling agent was 100 mM sucrose buffer with 20% (*v*/*v*) glycerol. Then, we applied a sinusoidal wave (10 V, 10 Hz) using a function generator at 60 °C in an incubator overnight and took out the sample from the incubator after formation. For pure PL GUV formation with DPPC, we added 0.1 mol% of 16:0 Liss Rhod PE to the chloroform stock solution and followed the same procedure except the formation time which was 1 day.

### 2.4. Giant Hybrid Unilamellar Vesicle (GHUV) formation by the PAPYRUS Method

For PL–BCP GHUV formation, we used the PAPYRUS method [29,30]. A 10 μL solution mixture in 25 mg/mL chloroform was prepared by mixing 3.0 μL of DPPC, 6.9 μL of mPEG-*b*-PCL, and 0.1 μL of (mPEG-*b*-PCL)-FITC. A piece of glass filter paper (1 cm in diameter) was cut into half and soaked with 10 μL of the mixed solution (25 mg/mL in chloroform) and 30 μL of toluene. The solution was evaporated by blow-drying until dried, then the paper was kept in vacuum for 1.5 h. For the hydration step, 750 μL of MilliQ water was introduced to the paper and incubated at 60 °C for 19 h.

### 2.5. Optical Microscopy Experiments

For Confocal Laser Scanning Microscopy (CLSM) and Differential Interference Contrast (DIC) imaging of GUVs and GHUVs, we used a LSM 800 confocal microscope (Carl Zeiss Microimaging GmbH, Germany). Giant vesicles were transferred to a coverslip (No. 1.5) before imaging. Images were processed in ImageJ (National Institutes of Health) including a 3D reconstructed figure [31]. Two sets of filters were used for detecting signals from FITC (excitation peak at 495 nm/emission peak at 519 nm) and Rhod B (excitation peak at 543 nm/emission peak at 565 nm). The CLSM images were obtained by reflection mode (RL) while the DIC image was produced by transmittance mode (TL).

### 2.6. Small Angle X-ray Scattering (SAXS) and Wide Angle X-ray Scattering (WAXS)

We carried out SAXS and WAXS by using a 13.3 keV X-ray beam at 12-ID-B beamline, Advanced Photon Source (APS), Argonne National Laboratory. Pilatus2M (Dectris) and Pilatus300 (Dectris) detectors with pixel sizes of 0.172 mm were used simultaneously for SAXS and WAXS, respectively. The sample to detector distance (SDD) was calibrated with a silver behenate powder standard. The SDD for SAXS and WAXS was 1998.81 mm and 455.26 mm, respectively. All samples were hydrated to a concentration of 100 mM of BCP, PL, or PL–BCP hybrid in quartz capillaries (Hilgenberg, Germany). Before the hydration step, samples were vacuum-dried for days to remove all organic solvents. We used MilliQ water for hydration. Quartz capillaries were flame sealed and kept hydrated for a couple of days before measurements. Experiments were all conducted at room temperature.

## 3. Results and Discussion

### 3.1. Pure mPEG-b-PCL BCPs and DPPC PLs form Giant Unilamellar Vesicles

Before studying a hybrid vesicle, it is essential to understand the self-assembly behavior of mPEG-*b*-PCL BCPs and PLs into pure component vesicles. Neat mPEG-*b*-PCL and DPPC bilayered vesicles are giant (several microns) versions of polymersomes and liposomes (hundreds of nm). Giant unilamellar vesicles (GUVs) were prepared by well-established electroformation methods [28] and investigated by CLSM. The amphiphilic structures of both mPEG-*b*-PCL and DPPC are illustrated in Figure 1. In mPEG-*b*-PCL diblock copolymer structure, methylated PEG (mPEG) is the hydrophilic block whereas PCL is the hydrophobic moiety. For DPPC lipid, phosphatidyl choline (PC) headgroup is the hydrophilic part and two 16:0 carbon chains are the hydrophobic moiety. As depicted in Figure 1, BCPs and PLs are analogously amphiphilic, yet liposomes and polymersomes exhibit distinct mechanical and physical properties [17]. One particular and important difference is the bilayer thickness of the self-assembled vesicles, which comes from their different molecular weights. Polymeric vesicles have an average bilayer thickness of 10–50 nm compared to lipid bilayers that are typically 3–5 nm thick [17]. The thickness mismatch of the BCP versus PL membranes plays a crucial role in the formation of hybrid PL–BCP membranes [8,9,14,16,32], in particular in molecular distribution and domain formation. Another factor that governs the morphology of BCP or PL vesicles is molecular packing or critical packing parameter (CPP), which is the ratio of the volumes occupied by the hydrophilic versus the hydrophobic blocks [1,33]. For BCPs this can be estimated by the weight fraction of the hydrophilic block, *f*_PEG_, and for mPEG-*b*-PCL (2k-*b*-5k g/mol) *f*_PEG_ = 0.29. The packing parameter of mPEG-*b*-PCL and DPPC [1,34] is consistent with the formation of bilayers that often close into vesicles [2,20,35,36]. Indeed, bilayers of mPEG-*b*-PCL [21,23] and DPPC [37,38] have shown to form vesicles of various sizes.

In order to directly observe GUVs and study their morphology by CLSM, we needed to synthesize appropriate fluorescence labels (Figure 2). For BCPs, we covalently linked fluorescein isothiocyanate (FITC) to the polymer hydrophobic core. Fluorescently labelled mPEG-*b*-PCL BCPs, or (mPEG-*b*-PCL)-FITC, were synthesized by conjugation chemistry of the isothiocyanate reactive group (-N=C=S) with the hydroxyl group at the end of PCL block accordingly to well-established protocols [27] (Figure 2a). For PLs we used commercially available 1,2-dipalmitoyl-sn-glycero-3-phosphoethanolamine-N-(lissamine rhodamine B sulfonyl) (ammonium salt)—16:0 Liss Rhod PE or Rhod B PE (Figure 2b). BCP and PL vesicles were labelled harnessing preferential secondary bonding interactions of trace amounts of fluorescent BCPs and PLs within the BCP and PL bilayer cores (Figure 2c).

Figure 3a shows a 2D cross-sectional CLSM image of a DPPC unilamellar vesicle which is about 15 μm in diameter. This is consistent with previously reported DPPC micron-scale vesicles—GUVs [38]. Although there has been a reasonable amount of studies on PL-GUVs systems [39,40,41], reports on BCP-GUV systems are comparatively scarce but have been gaining attention since the early 2010s [17]. Due to the fact that PEG and PCL polymers are biocompatible and approved by the Federal Drug Administration (FDA), most studies on these polymers and their ability to form nanoscale vesicles (polymersomes) have been focused on controlling or improving drug delivery efficiency for biomedical applications [20,42]. Figure 3b shows a CLSM image of cross section of a 10 μm in diameter mPEG-*b*-PCL vesicle. Appendix A show the three-dimensional Z-stack images of BCP-GUVs showing that the vesicles are mostly spheroidal and unilamellar. To our knowledge, vesicle formation using mPEG-*b*-PCL (2k-*b*-5k g/mol) in particular has not been reported, but this is consistent with similar BCP materials such as mPEG(2k)-*b*-PCL(13.5k) and mPEG(2k)-*b*-PCL(6k) that form unilamellar GUVs of 10–30 μm diameter [22]. Polymers with comparable molecular weight and *f*_PEG_ (PEG(1.1k)-*b*-PCL(2.9k), PEG(2k)-*b*-PCL(7.4k), PEG(5k)-*b*-PCL(10k), and PEG(5k)-*b*-PCL(16k)) have been shown to assemble into microspheres and irregularly shaped micron-scale vesicles from a few μm to around 30 μm in diameter [23]. In the mPEG-*b*-PCL BCP-GUVs of Figure 3b, we observe analogous irregular membranes with angular sections when compared to the smooth PL-GUV membranes. This could indicate that the hydrophobic core of the GUV bilayers is made of PCL experiencing packing frustration of its crystalline domains. The formation of smooth membrane GUVs seems to be energetically more favorable for PLs compared to BCPs. Both amphiphilic molecules are preferentially organized in a flat bilayer and the ability to form a curved membrane into a micro-scale vesicle will depend on the bending rigidity of the bilayer. While different PL and BCP systems yield diverse bending moduli, it is pretty generalizable that BCP membranes have bending moduli (κ_B_) significantly higher (ranging from tens to a few thousands k_B_*T*) than PLs (ranging from one to a few hundred k_B_*T*) [17,19]. This is the case even if one considers PL bilayers where lipids are in the gel-state, such as DPPC, compared to amorphous BCPs with very low glass transition temperature (*T*_g_) [17,26,43]. Notably, the κ_B_ obtained for crystalline BCP vesicles (denoted as aforementioned crystalsome, comprising PLLA-b-PEG BCP), was reported to be approximately 4000 k_B_T in aqueous solution [19]. This is remarkable because the thickness of the crystalsome membrane is smaller (ca. 4.5 nm) than typical BCP membrane (10–50 nm), yet the bending moduli greatly exceeds that of other BCPs. DPPC is known to have a κ_B_ of ˜275 k_B_*T* [43] and we can approximate that the bending modulus of the mPEG-*b*-PCL membrane would be similar to κ_B_ ˜4000 k_B_*T* due to the similar molecular structure and packing behavior between PCL and PLLA. We expected that a membrane with such a high bending modulus would be difficult to spontaneously close into a vesicle. However, previous works by the Discher group using mPEG(2k)-*b*-PCL(13.5k) and mPEG(2k)-*b*-PCL(6k) [22], and by the Therien group on (PEG(1.1k)-*b*-PCL(2.9k), PEG(2k)-*b*-PCL(7.4k), PEG(5k)-*b*-PCL(10k), and PEG(5k)-*b*-PCL(16k)) [23] show that vesicles can be formed by simple film hydration methods. Nonetheless, in these pure PL or BCP systems we used electroformation methods which are known to aid GUV synthesis for some challenging amphiphiles [17,44].

### 3.2. Giant Hybrid Unilamellar Vesicles (GHUVs) of mPEG-b-PCL and DPPC

Hybrid PL–BCP vesicles were prepared by the PAPYRUS method [29,30] in MilliQ water, which is an extension of the natural hydration method to prepare small unilamellar PL or BCP vesicles. The PAPYRUS method to prepare GUVs is important because it allows BCPs and PLs to mix in a monomeric state in an organic solvent prior to the hydration step where then BCP and PL molecules have the ability to self-organize into hybrid membranes or to completely segregate into sperate vesicular forms, depending on which configuration is energetically more favorable. The hybrid BCP–PL system comprises 20 mol% of mPEG-*b*-PCL (80 mol% DPPC) which is equivalent to 29.5 wt% DPPC. The BCP moieties are tagged with (mPEG-*b*-PCL)-FITC and the PL region with 16:0 Liss Rhod PE. As described in the previous section, PL membranes are more likely to bend into smooth GUV membranes when compared to BCPs, and for this reason we investigate a system comprising 20 mol% of mPEG-*b*-PCL and 80 mol% DPPC.

Figure 4 illustrates that DPPC and mPEG-*b*-PCL prefer to self-organize into a hybrid giant unilamellar vesicle instead of completely partition into PL-only or BCP-only GUVs. The DPPC–PEG-*b*-PCL GHUV adopts a diameter of ~20 μm and the membrane is smooth like that observed in DPPC GUVs. It is noteworthy that this is the first direct observation of the formation of a hybrid bilayer comprising mPEG-*b*-PCL and DPPC bent into a micron-scale vesicle. Two-dimensional cross-sectional CLSM images are shown in Figure 4a as well as a graphical illustration. The BCP domains are imaged in green at RL-FITC channel and the PL sites appear magenta at RL-Rhod B channel. It is clear that the giant vesicle bilayer comprises three distinct domains: PL-rich (mostly magenta), BCP-rich (mostly green), and mixed PL–BCP (green co-localized with magenta appearing mostly white). Figure 4b shows a 1D profile of the fluorescence intensity of each channel along the membrane of the vesicle depicted in Figure 4a. The intensity of the green channel is mostly constant along the vesicle rim indicating that BCP is present throughout the bilayer. However, the fluorescence intensity of the magenta channel fluctuates more prominently being significantly more intense in some regions indicating the presence of domains where PLs are preferentially clustered. When BCPs and PLs mix molecularly or within small nanoscale domains the fluorescence intensity of both channels is balanced and appears mostly white. Of course, these observations are simply qualitative, and the fluorescence intensity depends on the intrinsic susceptibility of dyes to experience bleaching. However, the presence of BCP-rich and PL-rich phase separated domains in other hybrid giant vesicular systems appears qualitatively analogous.

The GHUVs are imaged at several Z-stack projections, 1 μm apart. Figure 4c shows a compilation of Z-stack projections showing the upper hemisphere of the DPPC–mPEG-*b*-PCL GHUV as well as a graphical illustration. Each image of the Z-stack is displayed in Appendix A. In this collection of images, it is readily visible that indeed BCP-rich and PL-rich micro-scale domains form at the vesicle membrane coexisting with regions of homogeneous mixing. Figure 4d shows the fluorescence intensity profiles of the green and magenta channels across the black arrow of CLSM image in Figure 4c. In the compiled image of several Z-stacks, the fluorescence intensity profile also supports the picture of a GHUV comprising phase separated domains at the micro-scale rich in mPEG-*b*-PCL and rich in DPPC. The interaction between PEG-*b*-PCL BCP_S_ and DPPC PLs has been recently investigated in systems where the weight fraction of PEG is bigger than PCL (*f*_PEG_ = 0.70 vs. *f*_PEG_ = 0.29 in our work). The hybrid system assembles into vesicles smaller than 1 μm in diameter termed chimeric nanovesicles. The main observation is that PEG-*b*-PCL exerts steric pressure when anchoring onto DPPC bilayers inducing the formation of smaller vesicles compared to pure DPPC liposome [24].

Giant hybrid vesicles of DPPC and triblock copolymers PEG-*b*-PCL-*b*-PEG were recently studied by CLSM and 2D ^1^H Nuclear Magnetic Resonance (NMR) and the main finding was that when BCPs anchor onto the DPPC membrane the triblock copolymer prefers a “U-shape” over a “I-shape” conformation since the PCL block cannot orient vertically across a pure DPPC membrane [26]. The “U-shape” conformation of the BCP places the two ends of PEG blocks folded onto the outer membrane leaflet and the “I-shape” consists of a stretched BCP molecule penetrating through the DPPC bilayer and placing each PEG domain in the outer and the inner leaflet. In addition, it was observed that anchoring the BCP to the DPPC GUV results in a higher degree of stretchability which is consistent with the general observation that the stretch moduli of BCP membranes is considerably lower than that of PL membranes [17].

In the work presented here, the fraction of PCL is higher than PEG and we can see that pure mPEG-*b*-PCL GUVs have rather irregular membrane rims which are significantly smoother after DPPC inclusion. This is indicative that the molecular ordering of the hydrophobic core of the pure BCP vesicular membranes changes upon addition of PLs. The DPPC–mPEG-*b*-PCL is very robust retaining its phase separation behavior and shape even when exposed to hypertonic media, as will be described in Section 3.4.

### 3.3. DPPC–mPEG-b-PCL Hybrid Membrane Structure and Molecular Order

Recently, a micellar system made of PEG(5k)-*b*-PLLA(6k), termed *crystalsome* was investigated by cryo-transmittance electron microscopy and X-ray diffraction. The micelle has a diameter of ~200 nm and it was demonstrated that the BCP displayed a high degree of crystallinity yet showing continuous lattice distortions in order to fit into the curved space [18,19]. The DPPC–mPEG-*b*-PCL giant vesicular system is considerably larger than crystalsome micelles but the GHUVs core is aqueous so molecular packing, in particular the tendency of BCP or PL moieties to crystallize, are confined to a 4–10 nm thick membrane. To understand the self-assembly and molecular organization of mPEG-*b*-PCL and DPPC within a hybrid membrane we performed structural studies using Small and Wide Angle X-ray scattering (SAXS/WAXS). While PL alkyl chain packing has been investigated in PL–BCP hybrid membranes before [6,12,14,15,16], in nearly all systems studied so far the BCP component remains amorphous. In this paper, we wanted to investigate the behavior of a PL–BCP hybrid membrane when the BCP component has the ability to crystallize. If one or more blocks are able to crystallize, the structure or assembled morphology of the BCPs is starkly different when compared to amorphous BCPs. For example, the aforementioned crystalsome made of PEG(5k)-*b*-PLLA(6k) appears to be a rough vesicle with distortions of the crystalline domains and grain boundaries arising from high bending rigidity of crystalline BCP [19]. The most favorable configuration is a result of the interplay between intermolecular interactions that lead to self-assembly and crystallization pathways [45]. Therefore, it is crucial to investigate the influence of BCP crystallinity in PL–BCP hybrid membrane behavior both at meso, nano and molecular scales.

DPPC–mPEG-*b*-PCL fully hydrated hybrid membranes at the mPEG-*b*-PCL molar compositions of 0, 20, 40, 50, 60, 80, 100 mol% were investigated by X-ray scattering in Small and Wide Angle and are presented in Figure 5. Figure 5a shows a *I* vs. *q* 1D SAXS profile of hydrated neat mPEG-b-PCL (denoted as BCP in the figure), neat DPPC, and their mixture (from bottom to top: 0, 20, 40, 50, 60, 80, 100 mol% mPEG-*b*-PCL). The SAXS data for pure mPEG-*b*-PCL and pure DPPC displays the expected SAXS pattern arising from periodic stacks of self-assembled bilayers in a multilamellar arrangement throughout the capillary sample as shown in a left cartoon in Figure 5c. A series of equally spaced Bragg peaks are consistent with a multilamellar phase and are marked as L^00n^ where *n* denotes the diffraction order. The interlayer spacing of the multilamellar structure, which is called *d*-spacing, can be calculated from *d* = 2πn/*q*_n_ where *q*_n_ corresponds to the scattering vector of the nth order diffraction peak. Based on this, the *d*-spacing of the multilamellar structure in neat BCP (mPEG-*b*-PCL) and neat PL (DPPC) are calculated to be 207 and 65 Å, respectively. When mixed in a hybrid membrane, both lamellar phases are still present. This has been observed for the first time in our laboratory using other PL–BCP systems and it arises from the fact that the BCP-rich and the PL-rich domains are correlated across layers in full registry [6,12,46]. The lamellar *d*-spacing of DPPC in the hybrid system remains at 65 Å and that of mPEG-*b*-PCL increases slightly to 223 Å. The *d*-spacing mismatch is more pronounced when compared to other DPPC–BCP hybrid systems, so the BCP-rich and PL-rich domains are expected to be rather large to alleviate the distortion and interfacial tension between domains. Hydrophobic mismatch in combination with specific interactions of hydrophilic moieties with the hydration layer [47] have been postulated to promote domain formation within pure PL bilayers having a liquid-ordered and liquid-disordered domain registry across multilamellar systems. Indeed, PEG is expected to have more freedom at the water–membrane interface once it is intercalated with the small phosphatidylcholine (PC) headgroups of DPPC. Mixing DPPC with mPEG-*b*-PCL would alleviate some of the PEG steric repulsion and that might be a driving force for PL and BCP mixing at the molecular or nanoscale. In summary, an interplay between hydrophobic mismatch and hydrophilic interactions determines domain formation and size. Indeed, as shown in Figure 4, GHUVs display regions of mixed BCPs and PLs coexisting with micro-scale phase separated domains.

Figure 5b shows the WAXS data that was simultaneously obtained with SAXS for the same samples as shown in Figure 5a. PCL can adopt a semi-crystalline state at room temperature displaying an orthorhombic unit cell of lattice parameters: *a* = 7.5 Å, *b* = 5 Å, and *c* = 17.3 Å. Two parallel PCL chains align in opposite directions along the *c*-axis in a P2_1_2_1_2_1_ space group (the PCL unit cell is drawn in Figure 5c, right panel) [48,49]. For samples containing mPEG-*b*-PCL, we can detect four peaks at *q* = 1.51, 1.55, 1.67, and 1.71 Å^−1^ that fit to a Lorentzian line (Appendix A; Lorentzian fitting described in Appendix A) and can be consistently assigned to the diffraction of planes (110), (111), (200), and (210) of a PCL orthorhombic unit cell. In addition, we can observe diffraction peaks at *q* = 1.48 and 1.52 Å^−1^ also fitting to a Lorentzian line (Appendix A; Lorentzian fitting described in Appendix A) that arise from the packing of DPPC alkyl chains in an pseudo-hexagonal unit cell (PL gel phase) with *a* = 8.46 and *b* = 4.71 Å (also represented in Figure 5c) [50]. In the gel phase—L_β’_ [50,51,52,53,54,55] the DPPC alkyl chains are tilted with respect to the bilayer normal. The *q*-value at 1.48 Å^−1^ arises from the (20) planes at distance *d*_20_ = 4.24 Å and the peak at 1.52 Å^−1^ comes from the combined diffraction of the {11} family of planes at distances (*d*_11_ and *d*_1-1_) = 4.13 Å. It is noteworthy that some PCL crystal diffraction peaks overlap with the diffraction arising from alkyl chain packing in the DPPC gel phase. However, both SAXS and WAXS are consistent with a regular DPPC lamellar in the gel state - L_β’_ that is maintained after hybridization with mPEG-*b*-PCL. The DPPC gel phase peak at 1.48 Å^−1^ is present in all PL–BCP hybrid samples (80, 60, and 50 mol% PL). This observation is in contrast with WAXS and solid-state NMR studies from our laboratory [6,12] obtained for hybrid DPPC–BCP membranes where BCP was amorphous PBD-*b*-PEO. In this case, the amorphous hydrophobic core of the BCP induced a significant degree of disorder and local mobility of the DPPC alkyl chains that appear more fluid-like than gel-like. This result is important because it shows that the crystallinity state of the BCP strongly influences the state of PL membrane fluidity and order which set the mechanical and permeability properties of membranes.

The BCP (mPEG-*b*-PCL) and the PL (DPPC) components of the hybrid membranes both have temperature-dependent phase behavior. The melting temperature of the PCL part of the BCP is 52 °C [21] while PEO remains amorphous at room temperature (see Figure 5b). DPPC hydrocarbon chains transition from a gel-phase (all-*trans* C-C bonds) to a liquid-disordered state at 41 °C [56] and at room temperature DPPC remains in the gel-phase (L_β_—Figure 5b). Hence, it is likely that the BCP-rich domains emerge first during the cooling process of the formation of DPPC–mPEG-*b*-PCL GHUVs. As the temperature is increased above the melting temperature of PCL, one could expect that the driving force for phase separation is less and the miscibility of DPPC and mPEG-*b*-PCL is beneficial at the molecular level, however there will still be a considerable hydrophobic mismatch between the PL and the BCL chains leading to the persistence of domains, but at the nanoscale only. We are pursuing such studies in a separate effort that goes beyond the scope of this paper. Herein, we report that phase separation is present at temperatures below the transition temperatures of pure BCL and PL systems.

### 3.4. DPPC–mPEG-b-PCL GHUVs in Hypertonic Media

We investigated the effect of media on the DPPC–mPEG-*b*-PCL GHUVs by exposing the GHUVs having a MilliQ water interior to 100 mM glucose buffer which is a hypertonic environment. Figure 6a–c display 2D cross-sectional CLSM images of DPPC–mPEG-*b*-PCL GHUVs in the RL-FITC channel, the RL-Rhod B channel, and a merged image of both. The (mPEG-*b*-PCL)-FITC dye seems to be partially soluble in the glucose buffer as the background surrounding the vesicle displays a weak green fluorescence background signal. The shape of the vesicle is a bit more irregular but the vesicles remain overall spherically shaped, as shown in the 2D cross-sectional images in Figure 6a–c and Appendix A. Hypertonic media can lead to osmotic shrinkage of GUV size [57,58] and change of vesicle shape [59]. The DPPC–mPEG-*b*-PCL GHUV retains its size and the shape is only partially deformed, however, the membrane appears not as smooth as in a balanced osmotic stress set-up. This arises in pure BCP vesicles due to packing frustration and high bending rigidity of crystalline PCL domains as shown in Figure 3. Inclusion of PL under balanced osmotic conditions alleviates some of these constraints and the GHUV membranes appear smooth (Figure 5). Interestingly, some of the PL-rich and BCP-rich domains previously observed seem more prominent, clustering at the surface of the vesicle. (Additional CLSM data is shown in Appendix A.) These domains are much more distinctive than ones described in Figure 4. The 1D profiles of the fluorescence intensity of each channel along the vesicle membrane are included in Figure 6d and are consistent with a membrane with phase-separated DPPC and mPEG-*b*-PCL domains. Domain enhancement under osmotic imbalance has been observed in lipid-only GUVs [58]. In this work, it was shown that the swelled state osmotic pressure and elevated membrane tension, due to the influx of water, promotes domain formation. It is possible that an analogous phenomenon is observed here with in the DPPC–mPEG-*b*-PCL GHUVs system. However, we cannot exclude the possibility that a hypertonic environment induces aggregation of residual DPPC and mPEG-*b*-PCL dissolved in the GHUV media and their absorption onto the vesicle surface. Indeed, we were able to observe the interaction and collision of the GHUVs with some particulates or debris in solution (Appendix A). Appendix A, which is a vertically scanned CLSM video of DPPC–mPEG-*b*-PCL GHUV in hypertonic media (80 mol% DPPC with respect to mPEG-*b*-PCL), displays a track of the GHUV colliding to some clustered particulates. We cannot distinguish between enhanced phase separation or aggregation induced by hypertonic media but it is noteworthy that the GHUVs mostly retain their shape and size under hypertonic media showing that they are mechanically more robust than pure PL GUVs. The DIC image in Figure 6e and the 1D profile (inset) show that DPPC–mPEG-*b*-PCL GHUVs have an aqueous interior.

## 4. Conclusions

In this work, we have characterized a giant hybrid unilamellar vesicle (GHUV) comprising a phospholipid (PL) in the gel phase and a semi-crystalline block copolymer (BCP). Hybrid membranes composed of DPPC and mPEG-*b*-PCL (80 mol% DPPC with respect to mPEG-*b*-PCL) fold into several tens of micrometer sized vesicles in an aqueous environment. The hybrid membrane displays regions of large domains phase-separated into PL-rich and BCP-rich as well as regions of very small domains or molecularly mixed PL and BCP. The formation and size of the domains are governed by an interplay between minimization of steric forces between PEG tails as well as packing of PCL semi-crystalline domains. PEG steric forces favor DPPC mixing or small domain formation due to the fact that small PC headgroups act as a spacer between PEG chains. However, PCL hydrophobic groups have the ability to crystallize and form bilayers that are considerably thicker than PL membranes, resulting in the formation of lager, microscale domains. Pure PL and BCP can form GUVs as well. GUVS of pure mPEG-*b*-PCL are irregularly shaped compared to the smooth membrane vesicles of pure DPPC GUVs. The hybrid giant unilamellar vesicles are smooth resembling pure DPPC and mostly retain their size and shape even under hypertonic conditions. X-ray scattering and diffraction reveal that indeed these systems form a hybrid lamellar phase with DPPC remaining in a gel-state and PCL displaying diffraction peaks consistent with the existence of crystalline domains.

## Figures and Tables

**Figure 1 polymers-12-01232-f001:**
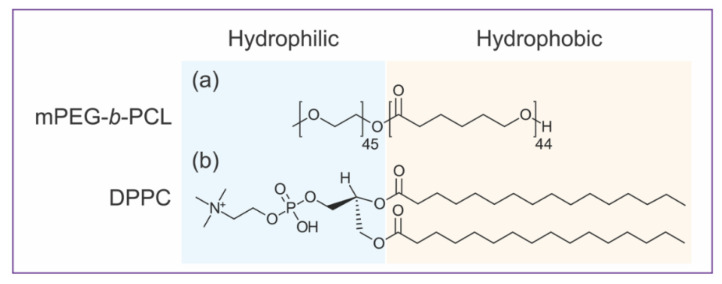
Amphiphilic structures of hybrid giant vesicles components: (**a**) Methoxy poly(ethylene glycol)-*block*-poly(ε-caprolactone) 2k-*b*-5k g/mol (mPEG-*b*-PCL) diblock copolymer (**b**) 1,2-dipalmitoyl-sn-glycero-3-phosphocholine (DPPC) lipid.

**Figure 2 polymers-12-01232-f002:**
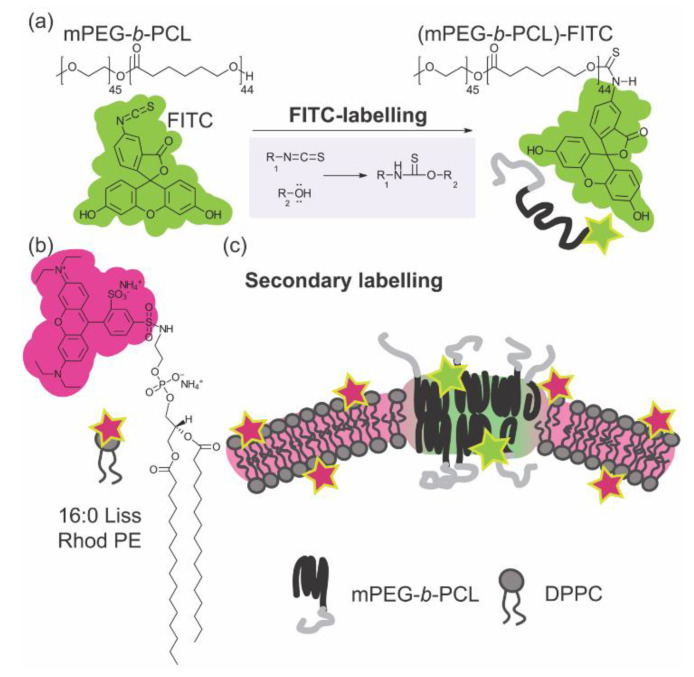
Scheme of (**a**) a fluorescent labelling chemistry of fluorescein isothiocyanate (FITC) with mPEG-*b*-PCL, (**b**) structure of 16:0 Liss Rhod PE (or Rhod B PE), and (**c**) secondary fluorescent labelling mechanism in DPPC–mPEG-*b*-PCL hybrid membrane using (mPEG-*b*-PCL)-FITC and 16:0 Liss Rhod PE.

**Figure 3 polymers-12-01232-f003:**
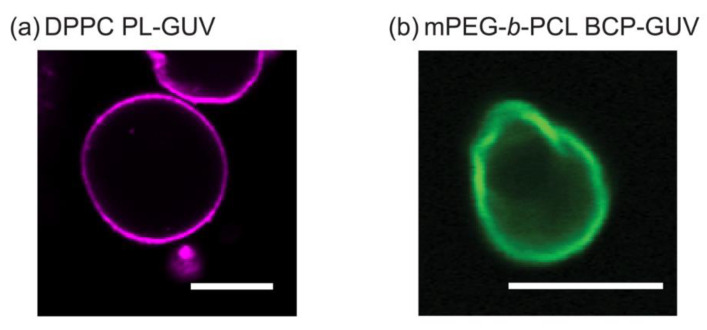
2D cross-sectional confocal laser scanning microscopy (CLSM) images of (**a**) DPPC giant unilamellar vesicle (GUV) (Scale bar: 20 μm) and (**b**) mPEG-*b*-PCL GUV (Scale bar: 10 μm). The substances 16:0 Liss Rhod PE and (mPEG-*b*-PCL)-FITC were used to tag phospholipid (PL)-GUVs and amphiphilic block copolymer (BCP)-GUVs, respectively. RL-Rhod B fluorescence in magenta (excitation laser at 561 nm/detection wavelength 550–700 nm) and RL-FITC fluorescence in green (excitation laser at 488 nm/detection wavelength 400–571 nm). Both images were obtained on a Plan-Apochromat 20 × /0.8 M27 objective lens. These GUVs encapsulated 100 mM sucrose buffer and were suspended in iso-osmolar solution of 100 mM glucose to lower the mobility during microscopic observation.

**Figure 4 polymers-12-01232-f004:**
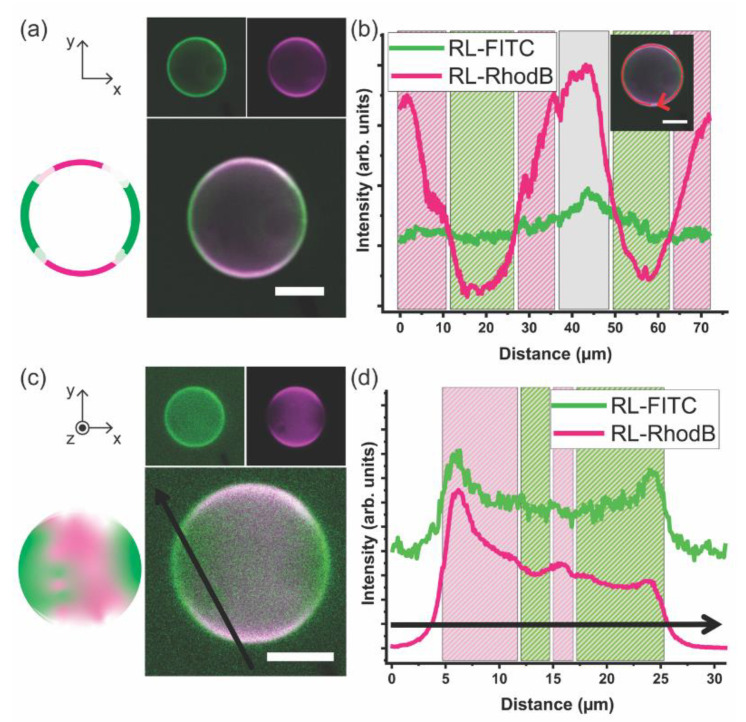
(**a**) 2D cross-sectional CLSM images of the DPPC–mPEG-*b*-PCL giant hybrid unilamellar vesicle (GHUV) (80 mol% DPPC and 20 mol% mPEG-*b*-PCL). (mPEG-*b*-PCL)-FITC and 16:0 Liss Rhod PE were used to tag BCP and PL domains, respectively. RL-FITC fluorescence in green (excitation laser at 488 nm/detection wavelength 400–544 nm) and RL-Rhod B fluorescence in magenta (excitation laser at 561 nm/detection wavelength 558–700 nm) are in separate channels on the top and the merged image of RL-FITC and RL-Rhod B channels is shown at the bottom. Left inset is a graphical illustration of the merged image. (**b**) 1D profile of each channel along the vesicle membrane in (**a**). (**c**) a Z-stack projection of height-resolved CLSM images of an upper hemisphere of the DPPC–mPEG-*b*-PCL GHUV (3D reconstruction). (mPEG-*b*-PCL)-FITC and 16:0 Liss Rhod PE were used to tag polymer domains and lipid domains, respectively. RL-FITC fluorescence in green (excitation laser at 488 nm/detection wavelength 400–544 nm) and RL-Rhod B fluorescence in magenta (excitation laser at 561 nm/detection wavelength 558–700 nm) are in separate channels on the top and the merged image of RL-FITC and RL-Rhod B channels is shown at the bottom. Left inset is a graphical illustration of the merged image. (**d**) 1D profile of each channel across the *Z*-axis projection image of the vesicle in (**c**). The internal and external medium of the vesicles is MilliQ water. Scale bar: 10 μm. Images were obtained on EC Plan-Neofluar 10 × /0.3 Ph 1 objective lens. Z-stack thickness: 1 μm.

**Figure 5 polymers-12-01232-f005:**
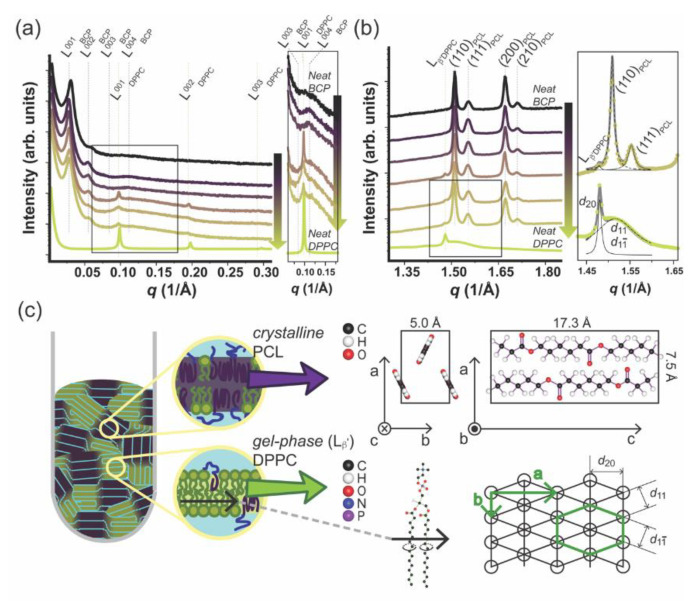
Simultaneous (**a**) Small Angle X-ray Scattering and (**b**) Wide Angle X-ray Scattering profiles obtained for hydrated neat mPEG-*b*-PCL(denoted as BCP above), neat DPPC, and their mixture. (From top to bottom; 0, 20, 40, 50, 60, 80, 100 mol% DPPC. Inset is a zoom-in image.) The function used for curve fitting is a Lorentz function (inset, (**b**)) (**c**) Graphical illustration of nanostructured hybrid material of mPEG-*b*-PCL and DPPC (left) and crystallographic information of each component (right).

**Figure 6 polymers-12-01232-f006:**
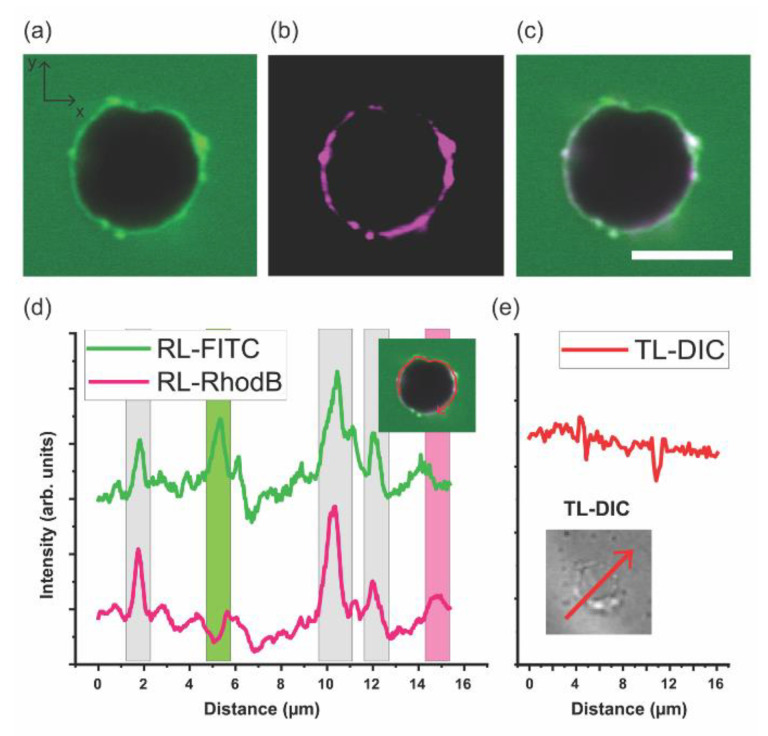
2D cross-sectional CLSM images of mPEG-*b*-PCL/DPPC GHUVs in hypertonic media (80 mol% DPPC with respect to mPEG-*b*-PCL). (mPEG-*b*-PCL)-FITC and 16:0 Liss Rhod PE were used to tag polymer domains and lipid domains, respectively. The vesicle encapsulates MilliQ water and the external dispersion medium is introduced with 100 mM glucose solution. Scale bar: 5 μm. (**a**) RL-FITC fluorescence in green (excitation laser at 488 nm/detection wavelength 400–544 nm) (**b**) RL-Rhod B fluorescence in magenta (excitation laser at 561 nm/detection wavelength 558–700 nm) (**c**) the merged image of both RL-FITC and RL-Rhod B channels (**d**) 1D profile of each channels along the vesicle membrane (**e**) TL-differential interference contrast (DIC) image (inset) and its 1D profile. All images were obtained on Plan-Apochromat 63 × /1.40 Oil DIC M27 objective lens.

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
