# Peer review of "Hybrid Unilamellar Vesicles of Phospholipids and Block Copolymers with Crystalline Domains"

_polymers, 2020, doi:10.3390/polym12061232_

Round 1

Reviewer 1 Report

The paper shows that vesicles formed from lipid and block copolymer mixture were more uniformed and stable than vesicles formed from a single component. The vesicles were characterized by a confocal microscope and x-ray scattering, which suggested that phase-separated vesicle structure with crystalline domains. The authors used only one microscope image to conclude that only phase-separated vesicles were formed. However, i think several images should be characterized to conclude that vesicles with a single component lipid only, block copolymer only were not formed. Also I only get a supporting movie, which is nothing to do with this paper. Thus, i recommend adding the statistics for the number of hybrid vesicles against single ones and true supporting information before publishing the paper. 

Author Response

Reviewer #1: The paper shows that vesicles formed from lipid and block copolymer mixture were more uniformed and stable than vesicles formed from a single component. The vesicles were characterized by a confocal microscope and x-ray scattering, which suggested that phase-separated vesicle structure with crystalline domains. The authors used only one microscope image to conclude that only phase-separated vesicles were formed. However, i think several images should be characterized to conclude that vesicles with a single component lipid only, block copolymer only were not formed. Also I only get a supporting movie, which is nothing to do with this paper. Thus, i recommend adding the statistics for the number of hybrid vesicles against single ones and true supporting information before publishing the paper.

We thank the reviewer for the comments. We are confident that the hybrid system comprises phase-separated domains rich in BCP and rich in PL because two very different techniques (confocal microscopy and x-ray scattering) are completely consistent with that interpretation. However, the reviewer brings up a good point. Confocal microscopy is a poor sampling technique and phase-separation needs to be assessed at a single-vesicle level. The sample needs to be dilute with just a few neighboring vesicles. There could be pure BCP and PL vesicles in a more concentrated sample. We take confidence, however, that the formation of hybrid vesicles is not a rare event as it was observed in a variety of different samples (Figure 4, Figure 6, Figure S4, Video S1, and Figure S5) and in more than one vesicle at a given dilute sample. We now revised the Supplementary Information (SI) document that by mistake was missing SI figures. Finally, while confocal microscopy is a poor sampling technique, X-ray is a bulk method that averages over many hybrid membranes and it is clear that BCP and PL do hybridize into a phase-separated membrane.

With respect to the supporting Video S1, we now include a description of it in the main manuscript (line 429-431) that reads: “Video S1, which is a vertically scanned CLSM video of DPPC - mPEG-b-PCL GHUV in hypertonic media (80 mol% DPPC with respect to mPEG-b-PCL), displays a track of the GHUV colliding to some clustered particulates”. This video is included in this paper because we have to address a possibility of aggregation of residual DPPC and mPEG-b-PCL dissolved in the GHUV media and their absorption onto the vesicle surface due to a hypertonic environment, rather than just claiming that an osmotic imbalance induces phase separation of DPPC and mPEG-b-PCL domains

Reviewer 2 Report

In this manuscript, Cecilia Leal and co-workers synthetized a hybrid vesicles consisted of DPPC phospholipids and a diblock copolymer (BCP, PEG-b-PCL). By adjusting the fraction of PCL block, the authors show that the PCL block crystalized in the membrane that promotes the segregation of DPPC and BCP. Using SAXS and WAXS techniques, the authors further investigate the interplay between hydrophobic mismatch and hydrophilic interactions in the membranes which gives rise to DPPC rich, BCP rich and homogenous domains.

Overall, I found this work is well design. The results are interesting to polymer science community and presented in a clear fashion. So. I’d like to recommend the paper to be published on Polymers given that the author can address the minor comment below.

  1. The phase behaviors of BCP and PCL crystallization are sensitive to changes in temperature. The author might want to comment on the effect of temperature on the the phase separation of the BCP and PL in their systems

Author Response

Reviewer #2: In this manuscript, Cecilia Leal and co-workers synthetized a hybrid vesicles consisted of DPPC phospholipids and a diblock copolymer (BCP, PEG-b-PCL). By adjusting the fraction of PCL block, the authors show that the PCL block crystalized in the membrane that promotes the segregation of DPPC and BCP. Using SAXS and WAXS techniques, the authors further investigate the interplay between hydrophobic mismatch and hydrophilic interactions in the membranes which gives rise to DPPC rich, BCP rich and homogenous domains.

Overall, I found this work is well design. The results are interesting to polymer science community and presented in a clear fashion. So. I’d like to recommend the paper to be published on Polymers given that the author can address the minor comment below.

We thank the reviewer for the positive assessment of the manuscript. In this revised version we address all the important concerns raised up by both reviewers.

  1. The phase behaviors of BCP and PCL crystallization are sensitive to changes in temperature. The author might want to comment on the effect of temperature on the phase separation of the BCP and PL in their systems  

We thank the reviewer for raising this very relevant point. Indeed, the reason we selected this particular polymer system is because we wanted to investigate the effect of temperature on the phase-behavior. Another manuscript focusing on that aspect is now being prepared as a separate communication. To clarify this point we modified the manuscript in line 398-410 that reads as follows: “The BCP (mPEG-b-PCL) and the PL (DPPC) components of the hybrid membranes both have temperature-dependent phase behavior. The melting temperature of the PCL part of the BCP is 52 C [21] while PEO remains amorphous at room temperature (see Figure 5b). DPPC hydrocarbon chains transition from a gel-phase (all-trans C-C bonds) to a liquid-disordered state at 41 C [46] and at room temperature DPPC remains in the gel-phase (Lβ' – Figure 5b). Hence, it is likely that the BCP-rich domains emerge first during the cooling process of the formation of DPPC - mPEG-b-PCL GHUVs. As the temperature is increased above the melting temperature of PCL, one could expect that the driving force for phase separation is less and miscibility of DPPC and mPEG-b-PCL to be beneficial at the molecular level, however there will still be a considerable hydrophobic mismatch between the PL and the BCL chains leading to the persistence of domains, but at the nanoscale only. We are pursuing such studies in a separate effort that goes beyond the scope of this paper. In here we report that phase separation is present at temperatures below the transition temperatures of pure BCL and PL systems”.

In connection with this added paragraph we include a new reference [46]. References [21] and [46] are the following: [21] Ghoroghchian, P.P.; Li, G.; Levine, D.H.; Davis, K.P.; Bates, F.S.; Hammer, D.A.; Therien, M.J. Bioresorbable vesicles formed through spontaneous self-assembly of amphiphilic poly(ethylene oxide)-block-polycaprolactone. Macromolecules 2006, 39, 1673–1675.

[46] Arias, J.M.; Tuttolomondo, M.E.; Díaz, S.B.; Ben Altabef, A. Reorganization of Hydration Water of DPPC Multilamellar Vesicles Induced by l -Cysteine Interaction. J. Phys. Chem. B 2018, 122, 5193–5204.